# Is Segmental Ureterectomy Associated with Inferior Survival for Localized Upper-Tract Urothelial Carcinoma of the Ureter Compared to Radical Nephroureterectomy?

**DOI:** 10.3390/cancers15051373

**Published:** 2023-02-21

**Authors:** Marco Paciotti, Khalid Y. Alkhatib, David-Dan Nguyen, Kendrick Yim, Stuart R. Lipsitz, Matthew Mossanen, Paolo Casale, Phillip M. Pierorazio, Adam S. Kibel, Quoc-Dien Trinh, Nicoló Maria Buffi, Giovanni Lughezzani, Alexander P. Cole

**Affiliations:** 1Center for Surgery and Public Health, Brigham and Women’s Hospital, Harvard Medical School, Boston, MA 02120, USA; 2Division of Urological Surgery, Brigham and Women’s Hospital, Harvard Medical School, Boston, MA 02115, USA; 3Department of Urology, IRCCS Humanitas Research Hospital, Rozzano, 20089 Milan, Italy; 4Department of Biomedical Sciences, Humanitas University, Pieve Emanuele, 20090 Milan, Italy; 5Division of Urology, University of Pennsylvania, Philadelphia, PA 19104, USA

**Keywords:** upper-tract urothelial carcinoma, ureter cancer, comparative effectiveness, segmental ureterectomy, nephroureterectomy

## Abstract

**Simple Summary:**

National cancer registry data showed that segmental ureterectomy is not associated with inferior survival compared to radical nephroureterectomy in upper-tract urothelial carcinoma patients. Segmental ureterectomy provides a valid surgical approach that does not meaningfully sacrifice survival outcomes within appropriately selected patients of upper urothelial carcinoma of the ureter. Given the potential renal functional preservation benefits, segmental ureterectomy should be considered for selected patients, especially if there is an increased probability of requiring adjuvant chemotherapy.

**Abstract:**

Segmental ureterectomy (SU) is an alternative to radical nephroureterectomy (RNU) in the treatment of upper-tract urothelial carcinoma (UTUC) of the ureter. SU generally preserves renal function, at the expense of less intensive cancer control. We aim to assess whether SU is associated with inferior survival compared to RNU. Using the National Cancer Database (NCDB), we identified patients diagnosed with localized UTUC of the ureter between 2004–2015. We used a propensity-score-overlap-weighted (PSOW) multivariable survival model to compare survival following SU vs. RNU. PSOW-adjusted Kaplan–Meier curves were generated and we performed a non-inferiority test of overall survival. A population of 13,061 individuals with UTUC of the ureter receiving either SU or RNU was identified; of these, 9016 underwent RNU and 4045 SU. Factors associated with decreased likelihood of receiving SU were female gender (OR, 0.81; 95% CI, 0.75–0.88; *p* < 0.001), advanced clinical T stage (cT4) (OR, 0.51; 95% CI, 0.30–0.88; *p* = 0.015), and high-grade tumor (OR, 0.76; 95% CI, 0.67–0.86; *p* < 0.001). Age greater than 79 years was associated with increased probability of undergoing SU (OR, 1.18; 95% CI, 1.00–1.38; *p* = 0.047). There was no statistically significant difference in OS between SU and RNU (HR, 0.98; 95% CI, 0.93–1.04; *p* = 0.538). SU was not inferior to RNU in PSOW-adjusted Cox regression analysis (*p* < 0.001 for non-inferiority). In weighted cohorts of individuals with UTUC of the ureter, the use of SU was not associated with inferior survival compared to RNU. Urologists should continue to utilize SU in appropriately selected patients.

## 1. Introduction

Upper-tract urothelial carcinoma (UTUC) is a rare cancer that accounts for 5–10% of all urothelial carcinomas, with an estimated annual incidence of less than two cases per 100,000 in western countries [1,2,3]. Currently, radical nephroureterectomy (RNU) with bladder cuff excision is the standard of care in patients with high-risk upper-tract urothelial carcinoma (UTUC), while nephron-sparing treatments are preferred for patients with low-risk disease. Previous studies have demonstrated that RNU has a significant and enduring detrimental impact on renal function, putting the patient at risk of chronic kidney disease (CKD) and related sequelae [4,5,6]. The changes in renal function might affect eligibility for adjuvant cisplatin-based therapy in 30–60% of patients who were eligible for chemotherapy before surgery [4,5,7]. Lastly, given the advanced age of presentation and its associated comorbidities, many patients have contraindications to RNU [8]. Segmental ureterectomy (SU) can be utilized in selected cases as an alternative for tumors confined to the ureter. Although SU has been generally demonstrated to have superior renal functional outcomes without compromising oncologic control [9], concerns regarding selection criteria, oncological safety, and efficacy remain. Several studies have reported higher rates of local tumor recurrence after the kidney-sparing approach [10], including with segmental ureterectomy [11]. However, oncological safety data of SU have mainly been generated from small institutional databases, with underpowered analyses due to the small sample sizes.

In this setting, we sought to compare the OS of patients with localized UTUC based on whether they received SU vs. RNU with bladder cuff excision to better elucidate long-term survival outcomes associated with each surgical approach. We hypothesized that SU would be non-inferior to RNU in terms of OS recognizing that the NCDB may overcome some of the issues with sample size.

## 2. Materials and Methods

### 2.1. Data Source and Study Population

The National Cancer Database (NCDB) is a joint effort between the Commission on Cancer, the American Cancer Society, and the American College of Surgeons. It is a national cancer registry that compiles data for more than 1500 commission-accredited cancer programs in the United States [12]. With more than 30 million records across the country, NCDB is an essential source of standardized data for cancer surveillance providing standardized data on patients, hospitals, and therapies [13]. We captured all adult patients (>18 years old) with a primary diagnosis of urothelial carcinoma of the ureter (ICD-10-CM code C66) between January 2004 and December 2015. Patients with non-urothelial histology were excluded. In order to focus on patients with localized disease, we also excluded patients with clinical or pathologic evidence of nodal or distant metastases; moreover, we excluded patients with missing histology, follow-up, or local therapy information.

### 2.2. Exposure of Interest

The exposure of interest was the type of surgery. Specifically, patients undergoing SU were compared with those undergoing RNU with bladder cuff excision. We excluded patients subjected to other types of surgery or with missing information regarding local therapy.

### 2.3. Definition of the Variable of Interest

The outcome of this study was OS, defined as the duration of elapsed months from the date of diagnosis to the date of death or censor at last follow-up. Specifically, OS, calculated as the number of months between the date of diagnosis and the date on which the patient was last contacted or died, of patients undergoing SU and RNU was compared; moreover, we reported five-year OS of the two groups.

### 2.4. Covariates

Baseline characteristics included age at diagnosis, gender, race (defined as “White”, “Black”, or “other”), and year of diagnosis. The Charlson comorbidity index (classified as 0, 1, 2, or ≥3) provided by the NCDB was used to account for underlying disease status. We abstracted sociodemographic and hospital-level covariates including county-level median household income (defined as “high” (≥$48,000) or “low” (<$48,000)), residence type, county-level median education level (defined as “high” (<13% of adults in the patient’s zip code who did not graduate from high school) or “low” (≥13% of adults in the patient’s zip code who did not graduate from high school)), insurance status, facility type, facility county, and facility US region. We used the TNM classification system to categorize clinical tumor classification (≤T1, T2 T3, T4, or Tx), according to the American Joint Committee on Cancer, 8th Edition [14]. Other tumor-specific variables included tumor grade (defined as “low”, “high”, or “unknown”), tumor size (categorized as “<2 cm”, “>2 cm”, or “unknown”), and presence of lymph vascular invasion. We included chemotherapy as a treatment-related covariate.

### 2.5. Statistical Analyses

Means and standard deviations or median and interquartile ranges (IQRs) were reported for normally or nonnormally distributed continuous variables, respectively. Categorical variables were presented as frequencies and proportions. Statistical significance in differences between groups was assessed using a *t*-test for continuous variables and a chi-square test for categorical variables. To assess for balance between groups, we used the standardized-differences approach to compare observed covariates between RNU and SU patients [15]. This quantitative method allowed us to assess the balance in baseline characteristics between treatment groups. A standardized difference ≥10% for a given covariate indicated a significant imbalance.

Multivariable logistic regression analysis was performed to identify independent predictors for undergoing SU compared to RNU in the unweighted population. To account for potential selection bias, a propensity score-overlap weighting (PSOW) method was used to balance baseline characteristic differences between RNU and SU groups. Each patient was assigned a statistical weight proportional to the probability of that patient belonging to the opposite treatment group. PSOW improves the balance and precision at the baseline relative to other conventional propensity score methods [16,17]. We used the standardized differences approach to assess the balance of the covariates between groups after weighting.

PSOW-adjusted Kaplan–Meier curves were generated to compare overall survival (OS) between patients who received SU and those who received RNU. Equality of survivor functions was tested using the log-rank test and Cox test of equality in unweighted and weighted population, respectively; then, we performed a univariable Cox regression analysis to calculate the PSOW-adjusted hazard ratio (HR) of SU [18]. Finally, we ran a non-inferiority test with a priori upper-bound for inferiority set as HR > 1.1 with a one-sided alpha of 0.05. All analyses were performed using Stata IC 16.0 (Stata Corp LLC, College Station, TX, USA). An institutional review board waiver was obtained from the Brigham and Women’s Hospital.

## 3. Results

A total of 13,061 men and women diagnosed with localized urothelial carcinoma of the ureter were included in this study. Median follow-up was 38.1 (18.2–70.1) months, in which 9016 underwent RNU with bladder cuff excision, while 4045 underwent SU. A flowchart describing cohort selection is shown in Figure 1. A comparison between 11,049 (45.8%) patients with urothelial carcinoma of the ureter who did not meet the study criteria and 13,061 (54.2%) patients who were included in the study is reported in Appendix A.

### 3.1. Baseline Patient Characteristics and Predictors of Receiving Segmental Ureterectomy

Baseline characteristics of the study population pre-weighting are shown in Table 1. The mean age at diagnosis was 71.9 ± 10. Prior to weighting, most of the covariates had a standardized difference smaller than 10% between the two groups. In our multivariable logistic regression analysis, independent factors associated with the decreased likelihood of receiving SU were female gender (OR, 0.81; 95% CI, 0.75–0.88; *p* < 0.001), advanced clinical T stage (cT3: OR, 0.67; 95% CI, 0.56–0.80; *p* < 0.001; cT4: OR, 0.51; 95% CI, 0.30–0.88; *p* = 0.015), and high-grade tumor (OR, 0.76; 95% CI, 0.67–0.86; *p* < 0.001). Having an age at diagnosis greater than 79 years was associated with increased probability of undergoing SU (OR, 1.18; 95% CI, 1.00–1.38; *p* = 0.047). Remaining independent predictors of receiving SU are summarized in Table 2.

### 3.2. Patient Characteristics and Survival Analyses

After overlap weighting, the distribution of baseline patient and tumor characteristics were well balanced between the groups; indeed, all standardized differences of weighted comparisons indicated differences smaller than 0.001 in the observed covariates between the two groups (Table 1). The five-year OS in patients undergoing SU was 53.1% (95% CI: 51.3–54.8) and 52.6% (95% CI: 51.0–53.8). The total number of events (i.e., death) observed in the whole population was 6518 (1982 and 4536 in patients undergoing SU vs. RNU, respectively). The PSOW HR for SU is 0.98 (95% CI, 0.93–1.04; *p* = 0.538). The HR met the non-inferiority criterion with a threshold HR of 1.1 and a one-sided alpha of 0.05 (*p* < 0.001 for non-inferiority) rejecting the null hypothesis of SU conferring inferior survival compared RNU. The PSOW survival curves are shown in Figure 2. The PSOW-adjusted Cox regression analysis is shown in Table 3. The survival curves of patients who received SU and RNU according to the T stage group and the grade of the disease are shown in Figure 3 (unweighted population) and Appendix A (weighted population).

## 4. Discussion

In this retrospective analysis, we aimed to investigate whether SU is associated with inferior long-term survival compared to RNU. Among 13,061 men and women receiving extirpative surgery for localized UTUC with a median follow-up of 38.1 months, we found that OS following SU was non-inferior to RNU.

Current guidelines by the European Association of Urology for UTUC recommend kidney-sparing surgery such as SU for: (1) low-risk localized tumors, (2) high-risk distal ureteral tumors, or (3) patients with impaired renal function and/or solitary kidney. Ultimately, RNU is the standard for high-risk UTUC regardless of location [1]. Patients who undergo RNU may experience a mean decrease in the estimated glomerular filtration rate (eGFR) by 24% and are at a higher risk of renal insufficiency [5]. Consequently, other groups have investigated an expanding role for more conservative therapies, with SU becoming the most common alternative surgical procedure to RNU.

In this study, we compared the OS of patients who received SU vs. RNU for localized UTUC using NCDB and we identified demographic and clinical predictors of underdoing SU. Given the rarity of this disease, retrospective observational data provide a key perspective on real-world outcomes. This study included more than 13,000 men and women, of whom more than 4000 have undergone SU, and this group is the largest in terms of population size that compared SU vs. RNU. In order to minimize confounding, we used overlap weighting, a propensity score method that mimics important attributes of a randomized clinical trial, such as a clinically relevant target population, covariate balance, and precision [16]. This statistical technique improves baseline balance and precision in comparison to other methods such as standard inverse probability weighting or matching [16,17]. We reached a full balance between the study groups for a large number of covariates, including patient, hospital, and tumor level characteristics. After adjusting for confounding, we found no difference in OS between patients undergoing SU compared to those who received RNU. Our results are consistent with the most recent systematic review and meta-analysis. Indeed, that survey included 18 studies for a total of 4797 patients undergoing either RNU or SU and found no statistically significant difference in the five-year metastasis-free survival nor the cancer-specific survival between the two surgical interventions [19]. Similar conclusions were reached by other authors on retrospective data [20]. However, our study included the largest population and used a robust study design to compare the two procedures. We identified several factors associated with the utilization of SU vs. RNU, as expected tumor characteristics associated with the use of SU included low cT stage, low-grade pathology, and tumor size <2 cm; additionally, low education attainment and being treated at a non-academic institution were also associated with a lower probability of undergoing SU; on the other hand, we found that age >79 and male gender were more likely to receive SU compared to RNU. Altogether, our findings corroborate the idea that this is a complex surgery for an uncommon disease, and it should, therefore, be performed in high-volume centers on well-selected patients.

Due to NCDB limitations, we could not retrieve pre-operative and post-operative data on renal function; however, several studies have shown that nephron-sparing surgeries are associated with higher post-operative eGFR, and a lower risk of developing CKD [19,20,21,22]. Thus, kidney-sparing surgeries such as SU are indicated in patients with a functional or anatomical solitary kidney, bilateral disease, or severe renal insufficiency [23]. The superior renal functional outcomes of SU could not only ameliorate the downstream sequelae of chronic kidney disease and dialysis when compared to RNU, but also guarantee a better chance for patients to be eligible for platinum-based chemotherapy. This is of particular importance given the increasing role of peri-operative systemic therapy for UTUC, supported by high-quality evidence [24,25,26,27]. Given the survival benefit conferred by adjuvant chemotherapy and the superior renal functional outcomes conferred by kidney-sparing surgery, SU may maximize the chance of receiving downstream chemotherapy compared to RNU in those very well selected HR UTUC patients who are candidates for kidney-sparing surgery. Lastly, while pre-existing renal function may be a confounder, we would generally expect men and women with worse pre-operative renal function to receive nephron-sparing surgery. Because impaired renal function is likely associated with worse overall survival, this should bias our non-inferiority analysis towards the null hypothesis (inferior overall survival in the SU group).

While our analysis adjusted for clinical T stage, there are doubtless technical factors such as tumor location and prior surgeries which could tend to lead a surgeon to preferentially offer a SU versus an RNU. Some of these factors relate to patient selection and cannot be captured in this type of analysis; there will always be an element of surgical decision-making around whether to offer an RNU. Despite these limitations, our study confirms that in appropriately selected patients, SU confers no worse survival than RNU and should be offered to men and women who have tumors suitable for this approach. UTUC is usually a lethal disease with a bad prognosis, therefore it is crucial to avoid under-treatment. With that said, recent data have emphasized the importance of multimodality treatment and the combination of surgery and systemic therapy is often required [24,28]. Thus, the dual aims of nephron sparing and maximal disease excision need to be balanced, our paper supports that the judicious use of nephron-sparing approaches does not meaningfully sacrifice survival.

As a result of the advancements in the endoscopic diagnostic optic instruments, utilization of diagnostic upper-tract endoscopy for UTUC increased. Adaptation of improved diagnostic technologies led to better risk stratification of the disease, resulting in a trend towards more conservative therapy. Future advancements in the diagnosis of UTUC will allow for better patient selection for kidney-sparing surgeries to mitigate side-effects and risks associated with RNU.

Our study is not devoid of limitations. Firstly, our study is limited by its observational retrospective design and the inherent limitations associated with such study designs. Secondly, although we corrected for possible selection bias using PSOW, there may be underlying differences between the SU cohort and the RNU cohort that were not detected by the available covariates. We believe that many of these unobserved factors are non-differential between RU and SNU and may, in fact, bias towards the null hypothesis of inferior survival in SU (e.g., worse renal function or comorbidities in men or women who received SU). Thirdly, NCDB data is also limited in terms of granularity of relevant data. For example, we were not able to identify pre-operative hydronephrosis, location of the ureteral tumor (proximal or distal), focality of the tumor, and pre-operative and post-operative renal function, as well as smoking status. Finally, we had no information about the follow-up schedule, and we could not assess cancer recurrence and cancer-specific survival.

## 5. Conclusions

In appropriately selected UTUC of the ureter patients treated in a real-world setting, SU provides non-inferior OS compared to RNU. Given the potential renal functional benefits of SU and the increased probability of requiring adjuvant chemotherapy in UTUC, the judicious use of SU is likely a valid surgical approach which does not meaningfully sacrifice survival outcomes.

## Figures and Tables

**Figure 1 cancers-15-01373-f001:**
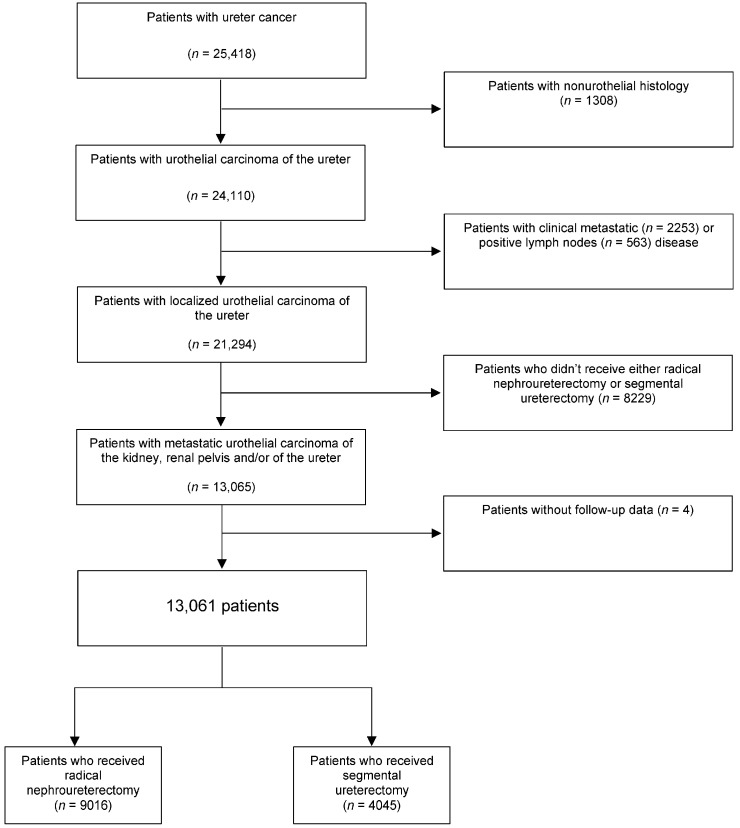
Flowchart that describes the selection of patients who received Radical Nephroureterectomy with bladder cuff excision (RNU) versus patients who received Segmental Ureterectomy (SU) for localized urothelial carcinoma of the ureter in the National Cancer Database, 2004 to 2015.

**Figure 2 cancers-15-01373-f002:**
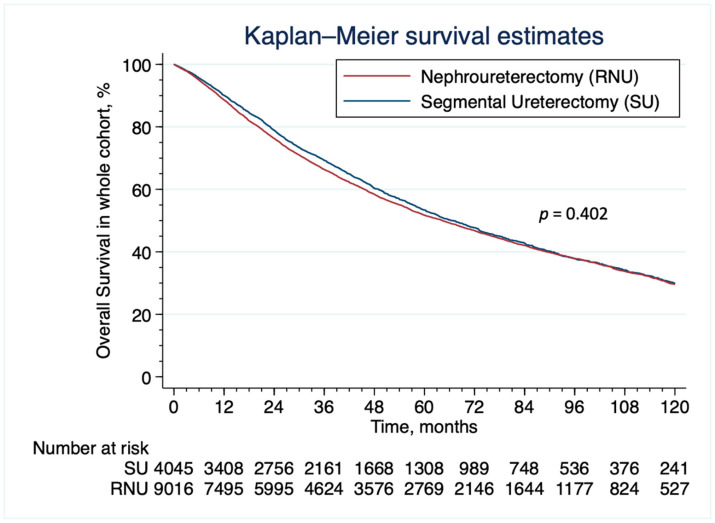
Overlap weighting-adjusted Kaplan–Meier analysis of overall survival in patients who received Radical Nephroureterectomy with bladder cuff excision vs. Segmental Ureterectomy for localized ureteral cancer. Number at risk from the non-adjusted population.

**Figure 3 cancers-15-01373-f003:**
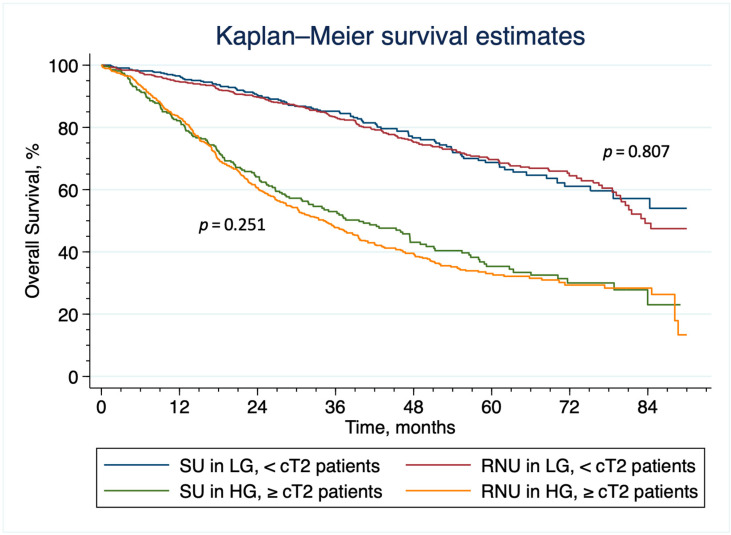
Kaplan–Meier analysis of overall survival in patients who received Radical Nephroureterectomy vs. Segmental Ureterectomy, according to grade and clinical T stage. SU= Segmental Ureterectomy, RNU= Radical Nephroureterectomy, LG= Low-Grade, HG= High-Grade, cT= Clinical T Stage.

**Table 1 cancers-15-01373-t001:** Baseline characteristics of patients treated with Radical Nephroureterectomy with bladder cuff vs. Segmental Ureterectomy for localized ureteral carcinoma in unweighted and weighted population.

Characteristic		Unweighted Population, No. (%)				Weighted Population, %			
		Overall *n* = 13,061	RNU *n* = 9016 (69%)	SU *n* = 4045 (31.0%)	Standardized Difference (%)	Overall	RNU	SU	Standardized Difference (%)
**Mean (SD) Age, Years**		71.9 (10.0)	71.8 (10.0)	71.9 (10.0)	1.1	71.9	71.9	72.0	0.8
Age categories, years	<60	1559 (11.9)	1078 (12)	481 (11.9)	−0.2	11.8	11.8	11.8	0
	60–69	3441 (26.4)	2371 (26.3)	1070 (26.5)	0.4	26.4	26.4	26.4	0
	70–79	4852 (37.2)	3379 (37.5)	1473 (36.4)	−2.2	36.8	36.8	36.8	0
	>80	3209 (24.6)	2188 (24.3)	1021 (25.2)	2.3	25.0	25.0	25.0	0
Gender, *n* (%)	Male	8239 (63.1)	5544 (61.5)	2695 (66.6)	**10.7**	65.0	65.0	65.0	0
	Female	4822 (36.9)	3472 (38.5)	1350 (33.4)	**−10.7**	35.0	35.0	35.0	0
Race, *n* (%)	White	12,104 (92.7)	8345 (92.6)	3759 (92.9)	1.4	92.8	92.8	92.8	0
	Black	443 (3.4)	314 (3.5)	129 (3.2)	−1.6	3.3	3.3	3.3	0
	Other	399 (3.1)	271 (3.0)	128 (3.2)	0.9	3.1	3.1	3.1	0
	Unknown	115 (0.9)	86 (1.0)	29 (0.7)	−2.6	0.8	0.8	0.8	0
Comorbidity index, *n* (%)	0	8377 (64.1)	5791 (64.2)	2586 (63.9)	−0.6	64.0	64.0	64.0	0
	1	3281 (25.1)	2253 (25.0)	1028 (25.4)	1.0	25.2	25.2	25.2	0
	2	1028 (7.9)	714 (7.9)	314 (7.8)	−0.6	7.9	7.9	7.9	0
	≥3	375 (2.9)	258 (2.9)	117 (2.9)	0.2	2.9	2.9	2.9	0
Insurance status, *n* (%)	Private	3218 (24.6)	2180 (24.2)	1038 (25.7)	3.4	25.1	25.1	25.1	0
	Medicaid	364 (2.8)	256 (2.8)	108 (2.7)	−1.0	2.7	2.7	2.7	0
	Medicare	9170 (70.2)	6351 (70.4)	2819 (69.7)	−1.6	70.1	70.1	70.1	0
	Uninsured	146 (1.1)	110 (1.2)	36 (0.9)	−3.2	1.0	1.0	1.0	0
	Unknown	163 (1.3)	119 (1.3)	44 (1.1)	−2.1	1.2	1.2	1.2	0
Income, *n* (%)	High	8054 (61.7)	5476 (60.7)	2578 (63.7)	6.2	62.6	62.6	62.6	0
	Low	4932 (37.8)	3494 (38.8)	1438 (35.6)	−6.6	36.8	36.8	36.8	0
	Unknown	75 (0.6)	46 (0.5)	29 (0.7)	2.6	0.6	0.6	0.6	0
Education, *n* (%)	High	8103 (62.0)	5491 (60.9)	2612 (64.6)	7.6	63.2	63.2	63.2	0
	Low	4891 (37.5)	3483 (38.6)	1408 (34.8)	−7.9	36.2	36.2	36.2	0
	Unknown	67 (0.5)	42 (0.5)	25 (0.6)	2.1	0.5	0.5	0.5	0
Facility type, *n* (%)	Academic	4563 (34.9)	2962 (32.9)	1601 (39.6)	**14.0**	37.1	37.1	37.1	0
	Non-academic	8467 (64.8)	6035 (64.9)	2432 (60.1)	**−14.2**	62.6	62.6	62.6	0
	Unknown	31 (0.2)	19 (0.2)	12 (0.3)	1.7	0.3	0.3	0.3	0
Facility location, *n* (%)	East	5766 (44.2)	4018 (44.6)	1748 (43.2)	−2.7	43.7	43.7	43.7	0
	Central	5546 (42.5)	3877 (43.0)	1669 (41.3)	−3.5	41.8	41.8	41.8	0
	West	1718 (13.2)	1102 (12.2)	616 (15.2)	8.7	14.3	14.3	14.3	0
	Unknown	31 (0.2)	19 (0.2)	12 (0.3)	1.7	0.3	0.3	0.3	0
Facility county (%)	Metro	10,531 (80.6)	7239 (80.3)	3292 (81.4)	2.8	80.9	80.9	80.9	0
	Urban	1928 (14.8)	1348 (15.0)	580 (14.3)	−1.7	14.7	14.7	14.7	0
	Rural	255 (2.0)	190 (2.1)	65 (1.6)	−3.7	1.7	1.7	1.7	0
	Unknown	347 (2.7)	239 (2.7)	108 (2.7)	0.1	2.7	2.7	2.7	0
Facility distance (%)	First	7158 (54.8)	5011 (55.6)	2147 (53.1)	−5.0	54.1	54.1	54.1	0
	Second	4194 (32.1)	2878 (31.9)	1316 (32.5)	1.3	32.2	32.2	32.2	0
	Third	1653 (12.7)	1094 (12.1)	559 (13.8)	5.0	13.1	13.1	13.1	0
	Unknown	56 (0.4)	33 (0.4)	23 (0.6)	3.0	0.5	0.5	0.5	0
Clinical T stage (%)	≤T1	5158 (39.5)	3454 (38.3)	1704 (42.1)	7.8	40.9	40.9	40.9	0
	T2	1079 (8.3)	741 (8.2)	338 (8.4)	0.5	8.3	8.3	8.3	0
	T3	827 (6.3)	640 (7.1)	338 (4.6)	**−10.6**	5.3	5.3	5.3	0
	T4	92 (0.7)	74 (0.8)	18 (0.4)	−4.7	0.5	0.5	0.5	0
	Unknown	5905 (45.2)	4107 (45.6)	1798 (44.5)	−2.2	45.0	45.0	45.0	0
Tumor size (%)	≤2 cm	3675 (28.1)	2258 (25.0)	1417 (35.0)	**21.9**	32.4	32.4	32.4	0
	>2 cm	6505 (49.8)	5065 (56.2)	1440 (35.6)	**−42.2**	41.8	41.8	41.8	0
	Unknown	2881 (22.1)	1693 (18.8)	1188 (29.4)	**25.0**	25.7	25.7	25.7	0
Tumor grade (%)	Low grade	1975 (15.1)	1291 (14.3)	684 (16.9)	7.1	15.8	15.8	15.8	0
	High grade	4392 (33.6)	3138 (34.8)	1254 (31.0)	−8.1	32.3	32.3	32.3	0
	Unknown	6694 (51.3)	4587 (50.9)	2107 (52.1)	2.4	51.8	51.8	51.8	0
Lymph vascular invasion (%)	Not present	4488 (34.4)	3160 (35.1)	1328 (32.8)	−4.7	33.6	33.6	33.6	0
	Present	954 (7.3)	693 (7.7)	261 (6.5)	−4.8	6.8	6.8	6.8	0
	Unknown	7619 (58.3)	5163 (57.3)	2456 (60.7)	7.0	59.5	59.5	59.5	0
Chemotherapy (%)	Not received	10,997 (84.2)	7530 (83.5)	3467 (85.7)	6.1	85.1	85.1	85.1	0
	Received	1568 (12.1)	1142 (12.7)	426 (10.5)	−6.7	11.1	11.1	11.1	0
	Unknown	496 (3.8)	344 (3.8)	152 (3.8)	−0.3	3.8	3.8	3.8	0

The standardized differences ≥10% are indicated in bold. The standardized difference is indicated as “0” when <0.1% and > −0.1%.

**Table 2 cancers-15-01373-t002:** Multivariable logistic regression analysis examining factors associated with undergoing segmental ureterectomy compared to radical nephroureterectomy.

Variable		Hazard Ratio	95% Confidence Interval	*p*-Value
Age	<60	Ref		
	60–69	1.06	0.92–1.22	0.433
	70–79	1.04	0.89–1.21	0.637
	>79	1.18	1.00–1.38	**0.047**
Gender	Male	Ref		
	Female	0.81	0.75–0.88	**<0.001**
Race	White	Ref		
	Black	1.00	0.81–1.25	0.970
	Other	0.990	0.79–1.24	0.922
	Unknown	0.76	0.49–1.17	0.209
Comorbidity index	0	Ref		
	1	1.09	1.00–1.19	0.061
	2	1.03	0.89–1.19	0.656
	≥3	1.06	0.84–1.33	0.632
Insurance status	Private	Ref		
	Medicaid	0.88	0.69–1.13	0.315
	Medicare	0.95	0.86–1.06	0.364
	Uninsured	0.76	0.51–1.13	0.171
	Unknown	0.79	0.55–1.15	0.217
Annual income	High	Ref		
	Low	0.95	0.86–1.05	0.297
	Unknown	2.87	0.68–1.21	0.150
Education	High	Ref		
	Low	0.89	0.81–0.98	**0.014**
	Unknown	0.15	0.02–1.28	0.083
Facility type	Academic	Ref		
	Non-academic	0.75	0.69–0.82	**<0.001**
	Unknown	1.39	0.66–2.95	0.386
Facility location	East	Ref		
	Central	1.02	0.94–1.12	0.570
	West	1.37	1.22–1.54	**<0.001**
Facility county	Metro	Ref		
	Urban	0.96	0.84–1.09	0.488
	Rural	0.77	0.57–1.05	0.098
	Unknown	0.97	0.75–1.25	0.807
Distance	First	Ref		
	Second	1.08	0.99–1.19	0.077
	Third	1.16	1.01–1.33	**0.040**
	Unknown	3.51	0.66–1.85	0.139
Year of diagnosis	2004/6	Ref		
	2007/9	1.08	0.97–1.21	0.157
	2010/12	1.27	0.99–1.61	0.056
	2013/15	1.18	0.92–1.51	0.190
Clinical T stage	≤ T1	Ref		
	T2	0.99	0.85–1.15	0.885
	T3	0.67	0.56–0.80	**<0.001**
	T4	0.51	0.30–0.88	**0.015**
	Unknown	0.93	0.86–1.02	0.120
Tumor grade	Low grade	Ref		
	High grade	0.76	0.67–0.86	**<0.001**
	Unknown	0.76	0.61–0.96	**0.023**
Tumor size	<2 cm	Ref		
	>2 cm	0.45	0.42–0.50	**<0.001**
	Unknown	1.12	1.01–1.24	**0.027**
Lymph vascular invasion	Not present	Ref		
	Present	1.11	0.94–1.31	0.239
	Unknown	1.32	1.15–1.51	**<0.001**

The *p*-values ≤ 0.05 are indicated in bold.

**Table 3 cancers-15-01373-t003:** Univariate Cox proportional hazards regression examining the risk of all-cause mortality in patients diagnosed with localized ureteral carcinoma after overlap weighting.

Heading		Hazard Ratio	95% Confidence Interval	*p*-Value
Type of Surgery	Radical Nephroureterectomy	1 [REF]	---	---
	Segmental Ureterectomy	0.98	0.93–1.04	0.538

## Data Availability

Restrictions apply to the availability of these data. Data were ob-tained from the National Cancer Database and are available from the authors with the permission of the National Cancer Database.

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
