# Peer review of "Is Segmental Ureterectomy Associated with Inferior Survival for Localized Upper-Tract Urothelial Carcinoma of the Ureter Compared to Radical Nephroureterectomy?"

_cancers, 2023, doi:10.3390/cancers15051373_

Round 1
Reviewer 1 Report
I thank the authors for doing a large-scale retrospective report on such an important topic. They found that segmental ureterectomy is not an inferior treatment modality for patients with localized urothelial carcinoma of the upper urinary tract.
Just a few comments are as follows:
1-Results, Fig 1: In the box, it was written as “….. metastatic in 13,065 patients”. However, it should be non-metastatic. Meanwhile, if the cancer is within the kidney, there shouldn't be a kidney in the box.
2-I couldn’t see the Kaplan Maier charts. Also, I could not see any table in the text despite Table 1 being given. Figures other than Figure 1 are also missing.
3-More comparisons could be made between the groups. For instance, I wonder about the survival of patients with pT2-3 disease in both RNU and SU groups. The same issue is also valid for low-risk and high-risk diseases.
Author Response
I thank the authors for doing a large-scale retrospective report on such an important topic. They found that segmental ureterectomy is not an inferior treatment modality for patients with localized urothelial carcinoma of the upper urinary tract.
Just a few comments are as follows:
1-Results, Fig 1: In the box, it was written as “….. metastatic in 13,065 patients”. However, it should be non-metastatic. Meanwhile, if the cancer is within the kidney, there shouldn't be a kidney in the box.
Thanks to the reviewer for noticing this mistake, that has been corrected.
2-I couldn’t see the Kaplan Maier charts. Also, I could not see any table in the text despite Table 1 being given. Figures other than Figure 1 are also missing.
We apologize for this error in formatting the manuscript. All the tables and figures have been included in the main document.
3-More comparisons could be made between the groups. For instance, I wonder about the survival of patients with pT2-3 disease in both RNU and SU groups. The same issue is also valid for low-risk and high-risk diseases.
We thank the reviewer for his/her suggestion.
We added a supplementary figure (Supp Fig 1) showing survival curves of patients undergoing RNU and SU, according to pTstage group. Unfortunately, the NCDB doesn’t allowed for a full risk-group classification. However, we added a second supp figure showing the survival curve of patients affected by low-grade and cT1 vs high-grade and cT2 or higher UTUC undergoing RNU vs SU.

Reviewer 2 Report
In this manuscript the authors are comparing factors associated with receipt of two different surgeries: segmental ureterectomy (SU) vs. radical nephroureterectomy (RNU), as well as overall survival between the two groups. The study utilizes data from 13,061 patients selected from the National Cancer Database, and the methods they use to compare the two groups are novel, particularly PSOW.
There are, however,some issues that the authors need to address to better understand the findings:
1. In the methods section the authors need to describe better the timing of bladder cancer diagnosis, timing of surgery and then follow-up after surgery. I am assuming that follow-up from treatment to overall survival is the time component used in survival analyses. This is described very briefly and needs to be elaborated.
2. Did the patients included in this study differ substantially from those who were not included? it might be useful to include a supplemental table where the authors compare demographic and clinical features of these two groups to understand better selection bias into the study.
3. The authors showed that there were several demographic and clinical characteristics associated with receipt of SU vs. RNU. I am assuming that these differences were accounted for when the authors applied PSOW in the Cox regression model. However, it might be useful to show standardized differences before and after the application of propensity score methods. A graph might be useful to show reduction in standardized differences.
4. It might be useful to examine also cancer-specific survival in addition to overall survival. That is an endpoint that is very important in cancer survival studies.
5. Did the authors have information about recurrence or progression of carcinoma during follow-up? It might be useful to compare these outcomes between the two surgical procedures groups.
Author Response
In this manuscript the authors are comparing factors associated with receipt of two different surgeries: segmental ureterectomy (SU) vs. radical nephroureterectomy (RNU), as well as overall survival between the two groups. The study utilizes data from 13,061 patients selected from the National Cancer Database, and the methods they use to compare the two groups are novel, particularly PSOW.
There are, however,some issues that the authors need to address to better understand the findings:
- In the methods section the authors need to describe better the timing of bladder cancer diagnosis, timing of surgery and then follow-up after surgery. I am assuming that follow-up from treatment to overall survival is the time component used in survival analyses. This is described very briefly and needs to be elaborated.
We thank the reviewer for his/her suggestion. The overall survival was calculated as the number of months between the date of diagnosis and the date on which the patient was last contacted or died. Since all the patients received surgery (either RNU or SU), we believe there was no risk of “immortal time bias”. We better described this in the methods, and we also added 5-year survival data. On the other hand, we do not have information about the follow-up schedule. We added this as a limitation of the study.
- Did the patients included in this study differ substantially from those who were not included? it might be useful to include a supplemental table where the authors compare demographic and clinical features of these two groups to understand better selection bias into the study.
We thank the reviewer for his/her suggestion. We added a supplemental table (Supp Table 1) showing the comparison of the baseline characteristics of patients with urothelial carcinoma of the ureter who were included in the study vs those who were not. A reference in the manuscript has been added (first paragraph of the Results).
- The authors showed that there were several demographic and clinical characteristics associated with receipt of SU vs. RNU. I am assuming that these differences were accounted for when the authors applied PSOW in the Cox regression model. However, it might be useful to show standardized differences before and after the application of propensity score methods. A graph might be useful to show reduction in standardized differences.
We thank the reviewer for pointing this out. We corrected a formatting error and included the table 1, showing the standardized differences in unweighted and weighted population, in the main file.
- It might be useful to examine also cancer-specific survival in addition to overall survival. That is an endpoint that is very important in cancer survival studies.
We agree with the reviewer that cancer-specific survival would be an important endpoint to assess. Unfortunately, the National Cancer Database (NCDB) does not include data on recurrence and cancer-specific survival. On the other hand, given the aggressiveness of the disease under investigation, overall survival might be an acceptable surrogate outcome for cancer-specific survival.
- Did the authors have information about recurrence or progression of carcinoma during follow-up? It might be useful to compare these outcomes between the two surgical procedures groups.
This would be indeed a valuable analysis but, unfortunately, the National Cancer Database (NCDB) does not include data on recurrence and cancer-specific survival.

Reviewer 3 Report
The authors have assessed whether SU is associated with inferior survival compared to RNU. Using the National Cancer Database. I have several comments and suggestions.
1. The inclusion and exclusion should be described more clearly.
2. Many important baseline characteristics such as tumor stage, tumor size, tumor grade, surgical approach, neoadjuvant and adjuvant therapy were not provided in the table 1.
3. As shown in Table 3, large tumor size, high tumor grade, and high grade are associated with poorer prognosis, could the authors provide sub-analyses of these groups of high-risk patients.
4. Besides OS, are there any other outcomes such as recurrence-free survival and cancer-specific survival?
5. Piraino JA et al also published a similar research in 2020 using the same database (Piraino JA, Snow ZA, Edwards DC, Hager S, McGreen BH, Diorio GJ. Nephroureterectomy vs. segmental ureterectomy of clinically localized, high-grade, urothelial carcinoma of the ureter: Practice patterns and outcomes. Urol Oncol. 2020 Nov;38(11):851.e1-851.e10. doi: 10.1016/j.urolonc.2020.08.004), the authors should discuss the novelty and strengths of this study.
Author Response
The authors have assessed whether SU is associated with inferior survival compared to RNU. Using the National Cancer Database. I have several comments and suggestions.
- The inclusion and exclusion should be described more clearly.
We thank the reviewer for his/her suggestion. That section has been improved.
- Many important baseline characteristics such as tumor stage, tumor size, tumor grade, surgical approach, neoadjuvant and adjuvant therapy were not provided in the table 1.
We corrected a formatting error and included the table 1 in the manuscript. The table includes comparative data about tumor stage, tumor size, tumor grade and chemotherapy. Unfortunately, data about the surgical approach have time limitation on the NCDB.
- As shown in Table 3, large tumor size, high tumor grade, and high grade are associated with poorer prognosis, could the authors provide sub-analyses of these groups of high-risk patients.
We thank the reviewer for noticing this. Unfortunately, the NCDB doesn’t allowed for a full risk-group classification. However, we added a second supp figure showing the survival curve of patients affected by low-grade and cT1 vs high-grade and cT2 or higher UTUC undergoing RNU vs SU. Please see also our reply to the reviewer #1.
- Besides OS, are there any other outcomes such as recurrence-free survival and cancer-specific survival?
Unfortunately, the National Cancer Database (NCDB) does not include data on recurrence and cancer-specific survival. Please see also our reply to the reviewer #2.
- Piraino JA et al also published a similar research in 2020 using the same database (Piraino JA, Snow ZA, Edwards DC, Hager S, McGreen BH, Diorio GJ. Nephroureterectomy vs. segmental ureterectomy of clinically localized, high-grade, urothelial carcinoma of the ureter: Practice patterns and outcomes. Urol Oncol. 2020 Nov;38(11):851.e1-851.e10. doi: 10.1016/j.urolonc.2020.08.004), the authors should discuss the novelty and strengths of this study.
We thank the reviewer for his suggestions. Differently from Piraino et al, we also included patients affected by low-grade UTUC. We believe that this population is very relevant considering the procedure under investigation. To the best of our knowledge, our study included the largest population among available publications comparing SU and NU. Moreover, we used what is considered the most robust PS method to balance the characteristics of the two groups. Considering that prospective and randomized studies are missing in this field, we believe that our studies can represent a valuable contribution to the literature body.
We added a couple of sentences to address this in the discussion.

Reviewer 4 Report
The findings of the present study are interesting for urologists. However, the results cannot be checked. It is probably an upload problem. Therefore, I recommend to re-submitt the manuscript after the revision of Tables.
Author Response
We apologize for this error in formatting the manuscript. All the tables and figures are included in the main document.
Round 2
Reviewer 2 Report
The authors have been responsive to reviewers' comments and the manuscript has improved. They also corrected a lot of issues with missing figures in prior version upload. Overall the study design and methods / statistical data analyses are sounds and the conclusions are appropriate.
There are, however, few minor issues to correct before considering for publication:
1. In Table 1 it might be useful to express standardized differences as percentages rather than absolute numbers. It gives the wrong impression that standardized differences before PS matching are very small, when in fact for some covariates they are 10% or higher.
1. in Table 1 and throughout manuscript it might be useful to indicate what is meant by low vs high education / what cut-off point was used?
2. Same point is also for low vs. high income although in one table the cutoff point of S48,000 was used. It might be useful to use some US National standards to make it more comparable to published literature
3. Please add number of events in Table 2 for each category? Some of these events counts are low so it is useful to indicate that.
4. In Supplemental Table 1 what do the authors mean by p>0.001? What is the exact p? Do they mean p<0.001??
5. The authors should add p-value from log-rank test for KM curves. As a suggestion, the KM curves stratified by LG vs HG in Suppl Figure 2 can be moved as main figure too.
Author Response
- In Table 1 it might be useful to express standardized differences as percentages rather than absolute numbers. It gives the wrong impression that standardized differences before PS matching are very small, when in fact for some covariates they are 10% or higher.
We thank the reviewer for his/her suggestion. We changed the table accordingly. - in Table 1 and throughout manuscript it might be useful to indicate what is meant by low vs high education / what cut-off point was used?
We thank the reviewer for his/her suggestion. This measure of educational attainment for each patient's area of residence is estimated by matching the zip code of the patient recorded at the time of diagnosis against files derived from the American Community Survey data. This item provides a measure of the number of adults in the patient's zip code who did not graduate from high school, and is categorized as equally proportioned quartiles among all US zip codes. The cut-off point was 13% adults who did not graduate from the high school.
The manuscript has been changed accordingly. - Same point is also for low vs. high income although in one table the cutoff point of S48,000 was used. It might be useful to use some US National standards to make it more comparable to published literature
Similarly, the median household income for each patient's area of residence is estimated by matching the zip code of the patient recorded at the time of diagnosis against files derived from the American Community Survey data. Household income is categorized as quartiles based on equally proportioned income ranges among all US zip codes.
Within the NCDB, the median household income is a categorical variable (Code Label 1 Less than $38,000; 2 $38,000 - $47,999; 3 $48,000 - $62,999; 4 $63,000 +; blank Not Available).
Consistently with the previous literature, we simplified the variable into another categorical variable (i.e., “Income”) codified as following: low-income <$48,000; high-income $48,000+; unkown).
We has specified the definition of this variable in the method section. - Please add number of events in Table 2 for each category? Some of these events counts are low so it is useful to indicate that.
We thank the reviewer for giving us the opportunity to clarify this. The number of cases for the different categories can be found in table 1. Given the size of the population, several thousands of patients are included in each category.
- In Supplemental Table 1 what do the authors mean by p>0.001? What is the exact p? Do they mean p<0.001??
We thank the reviewer for pointing out this typo, that has been corrected. - The authors should add p-value from log-rank test for KM curves. As a suggestion, the KM curves stratified by LG vs HG in Suppl Figure 2 can be moved as main figure too.
We thank the reviewer for his/her suggestion. The manuscript has been changed accordingly. Supp figure 2 is the new figure 3, and shows survival curves in the unweighted population (thanks to Reviewer #4’s advice).

Reviewer 4 Report
The present study showed the comparison of survival outcomes between segmental ureterectomy and radical nephrectomy in a larger cohort, using PSOW multivariable survival model. These findings are important for urologists. However, there are some concerns to be addressed in the present study.
1. Was the mean age at diagnosis was 71.8? (Page 5, Line 149). This is not consistent with Table 1.
2. The authors showed the PSOW-adjusted KM curves in Figure 2. However, no adjusted survival curves between segmental ureterectomy and radical nephrectomy is also important for clinicians.
Author Response
- Was the mean age at diagnosis was 71.8? (Page 5, Line 149). This is not consistent with Table 1.
We thank the reviewer for pointing out this typo, that has been corrected.
- The authors showed the PSOW-adjusted KM curves in Figure 2. However, no adjusted survival curves between segmental ureterectomy and radical nephrectomy is also important for clinicians.
We thank the reviewer for his suggestion. We changed Supp figure 2 (that is the new figure 3 following Reviewer #2’s advice), in order to show and test survival curves in the unweighted population.
